# Schistosoma haematobium infection is associated with lower serum cholesterol levels and improved lipid profile in overweight/obese individuals

Jeannot F. Zinsou[1,2], Jacqueline J. Janse[1], Yabo Y. Honpkehedji[1,2], Jean Claude Dejon-Agobé[2], Noemí García-Tardón[1], Pytsje T. Hoekstra[1], Marguerite Massinga-Loembe[2,5,6], Paul L. A. M. Corstjens[3], Govert J. van Dam[1], Martin Giera[4], Peter G. Kremsner[2,5], Maria Yazdanbakhsh[1], Ayola A. Adegnika[1,2,5,6], Bruno Guigas[1]*

1 Department of Parasitology, Leiden University Medical Center, Leiden, The Netherlands, 2 Centre de Recherches Médicales de Lambaréné, Lambaréné, Gabon, 3 Department of Cell and Chemical Biology, Leiden University Medical Center, Leiden, The Netherlands, 4 Center for Proteomics and Metabolomics, Leiden University Medical Center, Leiden, The Netherlands, 5 Institut für Tropenmedizin, Universität Tübingen, Tübingen, Germany, 6 German Center for Infection Research, Tübingen, Tübingen, Germany

* b.g.a.guigas@lumc.nl

**Data Availability Statement:** All relevant data are within the manuscript and its Supporting Information files.

## Abstract

Infection with parasitic helminths has been reported to improve insulin sensitivity and glucose homeostasis, lowering the risk for type 2 diabetes. However, little is known about its impact on whole-body lipid homeostasis, especially in obese individuals. For this purpose, a cross-sectional study was carried out in lean and overweight/obese adults residing in the Lambaréné region of Gabon, an area endemic for *Schistosoma haematobium*. Helminth infection status, peripheral blood immune cell counts, and serum metabolic and lipid/lipoprotein levels were analyzed. We found that urine *S. haematobium* egg-positive individuals exhibited lower serum total cholesterol (TC; 4.42 *vs* 4.01 mmol/L, adjusted mean difference [95%CI] -0.30 [-0.68,-0.06]; P = 0.109), high-density lipoprotein (HDL)-C (1.44 *vs* 1.12 mmol/L, -0.24 [-0.43,-0.06]; P = 0.009) and triglyceride (TG; 0.93 *vs* 0.72 mmol/L, -0.20 [-0.39,-0.03]; P = 0.022) levels than egg-negative individuals. However, when stratified according to body mass index, these effects were only observed in overweight/obese infected individuals. Similarly, significant negative correlations between the intensity of infection, assessed by serum circulating anodic antigen (CAA) concentrations, and TC (r = -0.555; P<0.001), HDL-C (r = -0.327; P = 0.068), LDL-C (r = -0.396; P = 0.025) and TG (r = -0.381; P = 0.032) levels were found in overweight/obese individuals but not in lean subjects. Quantitative lipidomic analysis showed that circulating levels of some lipid species associated with cholesterol-rich lipoprotein particles were also significantly reduced in overweight/obese infected individuals in an intensity-dependent manner. In conclusion, we reported that infection with *S. haematobium* is associated with improved lipid profile in overweight/obese individuals, a feature that might contribute reducing the risk of cardiometabolic diseases in such population.

**Funding:** This work was supported by funding from the Dutch Organization for Scientific Research: [ZonMw TOP Grant 91214131 to MY and BG] & [NWO project 184.034.019 to MG] https://www.zonmw.nl/nl/. The funders had no role in study design, data collection and analysis, decision to publish, or preparation of the manuscript.

**Competing interests:** The authors have declared that no competing interests exist.

## Author summary

Infection with parasitic helminths has been reported to be beneficial for metabolic homeostasis by improving insulin sensitivity and lowering the risk for developing type 2 diabetes. Elevated circulating cholesterol and triglyceride levels associated with obesity are also risk factors for cardiometabolic diseases. In the framework of a cross-sectional study conducted in an endemic rural area, we have investigated the impact of infection with *Schistosoma hematobium* on serum lipid homeostasis in adult individuals with a broad range of body weight. We found that helminth infection is associated with a lower serum total cholesterol (TC), high-density lipoprotein (HDL)-C and triglyceride (TG) levels, especially in overweight/obese individuals. Furthermore, significant negative correlations between the intensity of infection and TC, HDL-C, LDL-C and TG levels were also found in overweight/obese individuals but not in lean subjects. Altogether our study show for the first time that infection with *Schistosoma hematobium* is associated with an improved serum lipid profile in overweight/obese humans, a feature that may contribute to protection against cardiometabolic diseases in such population. Further investigation is however required to elucidate the underlying molecular mechanisms.

## Introduction

About one quarter of Earth inhabitants are infected with parasitic helminths and most of them are living in Africa [1]. In tropical and subtropical areas, soil-transmitted helminths (STH) are widely distributed, including *Ascaris lumbricoides*, *Trichuris trichiura* and hookworms, together with filarial nematodes and schistosomes such as *Schistosoma mansoni* and *Schistosoma haematobium* [1, 2]. Schistosomiasis affects 200 to 300 million people worldwide [1], especially in the poorest regions of the low and middle income countries (LMIC), and remains a major public health problem in these endemic areas [3, 4], where it co-exists with other health condition, including non-communicable diseases (NCDs).

NCDs such as obesity, metabolic syndrome and type 2 diabetes (T2D) are increasing not only in industrialized countries but also in developing countries that experienced rapid rural-urban transition, notably in Africa [5–7]. In 2017, 16 million African people were estimated to have T2D, with a projected doubling of this number in the coming 20 years [5]. Obesity is often associated with chronic low-grade inflammation, also called meta-inflammation, which contributes to peripheral and systemic insulin resistance, alterations of glucose and lipid metabolism, and ultimately leads to the development of T2D and related cardiovascular diseases [8]. Meta-inflammation mostly results from alterations in the composition and/or activation state of a variety of innate and adaptive immune cells in metabolic organs, notably in adipose tissue and the liver, creating a pro-inflammatory environment that contribute to tissue-specific insulin resistance and whole-body metabolic dysfunctions [9].

Chronic helminth infection is known to trigger a potent $T_H2$ immune response together with T-cell hyporesponsiveness through induction of a regulatory network [10], which have been both suggested to dampen meta-inflammation and restore tissue-specific and whole-body insulin sensitivity [9]. Interestingly, helminth infection and treatment with helminth-derived molecules have also been shown to improve insulin sensitivity and glucose homeostasis in various rodent models of T2D, at least partly through induction of a type 2 immune response in metabolic organs [11]. In line with this, epidemiological studies conducted in countries endemic for various helminth species have reported an inverse association between

helminth infection and insulin resistance [12] or incidence of metabolic syndrome and T2D [13, 14], suggesting a positive impact of helminths on host metabolic homeostasis. More recently, clinical interventional trials using anthelmintic drugs have also shown that deworming was associated with impaired systemic insulin sensitivity and glucose homeostasis in either STH-infected individuals or type 2 diabetic subjects with *Strongyloides stercoralis* infection, respectively [15, 16]. Nevertheless, the few human studies performed so far have mostly been done in STH-infected lean individuals, and principally focused on glucose rather than lipid homeostasis [11]. As such, little is currently known on the impact of other helminth infection, notably with schistosomes, on metabolic homeostasis in people with a broad range of body weight. We have therefore conducted a cross-sectional study to investigate whether infection with *S. haematobium* could affect glucose and lipid homeostasis in lean and obese/overweight adult individuals living in endemic rural area.

## Methods

### Ethics statement

The study was approved by the institutional ethics committee of the Centre de Recherches Médicales de Lambaréné (CERMEL), Lambaréné, Gabon (Registered number: CEI-MRU 002/2014) and was conducted in line with the Good Clinical Practice principles of the International Conference on Harmonization and the Declaration of Helsinki.

### Study site

The study was conducted in Lambaréné region, an endemic area for *S. haematobium* in Gabon. The schistosomiasis prevalence among children and young adults was recently found to be 26% in Lambaréné village (Jean Claude Dejon-Agobé, personal communication) and 45% in the surrounding Zilé-PK area [17, 18].

### Participants and study procedures

Adult individuals (>18 year-old) with either low/normal (<25) or high (>25) BMI and living in the study area were invited to join the study during on-site visit. An *a priori* power calculation was performed using mean serum TC levels from a previous small cohort study to determine the sample size allowing to detect a mean difference of ~12,5% between Sh⁻ and Sh⁺ group ($\alpha = 0.05$; power = 80%). The recruitment was carried out from July 2014 to April 2015. At screening, anthropometric measurements were performed according to the NHLBI practical guidelines (http://www.nhlbi.nih.gov). Eligible participants who provided urine, stool and blood samples were included for analysis. One sample of urine was collected during three consecutive days for the diagnosis of schistosomiasis. One stool sample was collected for assessing STH infection. Venous blood was collected between 8 and 10 am after an overnight-fast using S-Monovette EDTA tube (Sarstetd) for hematological analysis and BD Vacutainer SST II dry tubes (Beckton Dickinson) for serum biochemical analysis. Pregnant women (urinary β-HCG positive test), subjects with a family history of diabetes, non-diagnosed diabetic subjects (fasting serum glucose >7 mM), anyone positive for *Plasmodium falciparum* parasites, and any participants negative for the presence of *S. haematobium* eggs in urine but infected with at least one other helminths were excluded from the study (S1 Fig). At the end, 71 clinically healthy participants were included and two study groups were considered: a *S. haematobium* egg-positive (Sh⁺) group with all individuals clinically asymptomatic for schistosomiasis (n = 32), and a helminth negative (Sh⁻) group including participants negative for schistosomiasis and STH infection (n = 39). Individuals found positive for schistosomiasis (positive for

urine eggs) and/or for STH infections (positive for stool eggs) were later treated with a single dose of 40 mg/kg praziquantel and/or with 400 mg of albendazole daily for three consecutive days, respectively, and those infected with *P. falciparum* were treated with 80/480 mg artemether/lumefantrine tablet twice a day during 3 consecutive days.

## Parasitological examination

At inclusion, schistosomiasis was evaluated using the microscopic urine filtration method as described [17]. Participants were classified as *Sh*⁺ if at least one egg was detected, and as *Sh*⁻ if three consecutive urine screenings were negative. The infection with STH, including *Ascaris lumbricoides*, *Trichuris trichiura* and *Necator americanus* hookworms, was determined in fresh stool samples using the Kato-Katz technique [17]. A modified agar coproculture protocol was used for the detection of hookworms and *Strongyloides* larva. A thick blood smear was analyzed by microscopic examination for detecting *Plasmodium falciparum* parasites [17].

## Hematological and biochemical analysis

Hematological analysis was performed using an ABX Pentra 60 hematology system (Horiba Medical). For biochemical analysis, serum alanine aminotransferase (ALAT), aspartate aminotransferase (ASAT), glucose, total cholesterol (TC), high density lipoprotein-cholesterol (HDL-c) and triglycerides (TG) were measured on a Modular Analytics P-800 system (Roche Diagnostics). Low density lipoprotein-cholesterol (LDL-c) was calculated applying the Friedewald calculation. Serum insulin and C-peptide were measured on an Immulite 2500 automated system (Siemens Healthcare Diagnostics). High-sensitivity C-reactive protein (hsCRP) was measured using a multiplex array (Meso Scale Discovery). TIgE was measured by ELISA, as previously described [16]. HOMA-IR was calculated as fasting insulin (mU/L) x fasting glucose (mmol/L)/22.5.

## Circulating anodic antigen assay

Serum Circulating Anodic Antigen (CAA) concentrations were determined by the up-converting phosphor lateral flow (UCP-LF) assay as previously described [19]. Standards with known CAA concentrations were included to generate a calibration curve to calculate individual CAA levels and to validate the threshold of the assay; the maintained lower limit of detection was 10 pg/ml.

## Serum lipidomics

Lipids were extracted from 100 μL of serum by the methyl-tert-butylether method and analyzed using Lipidyzer, a direct infusion-tandem mass spectrometry (DI-MS/MS)-based platform (Sciex, Redwood City, California). Lipid concentrations were expressed as nmol/g of serum.

## Statistical analysis

The statistical analysis was carried out using SPSS 23.0 software package (SPSS Inc) or GraphPad prism 8.3 (GraphPad Software). Quantitative variables were reported as mean and standard deviation (SD) if normally distributed, otherwise they were expressed as median and interquartile range (IQR) and log-transformed for analysis. Unpaired *t*-tests were used to compare study groups. Adjustment for confounding factors was done by multivariate analysis using linear regression, and the differences between groups were expressed as mean and 95% confidence interval (CI). Odd ratios were calculated using multiple logistic regression analysis.

One-Way ANOVA with Tukey's post hoc test for multiple comparisons was used to compare the significant differences among the CAA stratified groups. Two-Way ANOVA with Tukey's *post hoc* test for multiple comparisons was used for lipidomic analyses of individual lipid species belonging to the same lipid class. *P* values <0.05 were considered to be statistically significant.

## Results

### Study population characteristics

The anthropometric characteristics of the study population and the hematological and serum biochemical parameters measured at inclusion are shown in Table 1. Based on urine egg count, 39 participants on a total of 71 individuals (55%) were found positive for *S. haematobium* and, among them, 7 were also co-infected with at least one STH species (17.9%). There were no differences in age, gender, BMI, and serum levels of hs-CRP, ALAT and ASAT between the two groups (Table 1), whereas a significant increase in both serum total IgE levels

**Table 1. Characteristics of the study population.**

| | *S. haematobium* negative (n = 32) | *S. haematobium* positive (n = 39) | *P*-value | Mean difference adjusted for age, sex, BMI and other helminths (95% CI) | *P*-value |
|---|---|---|---|---|---|
| **Age (year)** (mean, range) | **35.7** (18–63) | **34.5** (18–63) | 0.68 | | |
| **Male (%)** | **43.8** | **48.7** | | | |
| **BMI (kg/m$^2$)** (mean, SD) | **26.8** (6.9) | **25.6** (4.5) | 0.41 | | |
| *S. haematobium* **urine eggs** (median, IQR) | **0** (0–0) | **12** (4–63) | **<0.001** | | |
| **Other helminths (%)** | **0** | **17.9** | | | |
| *Ascaris lumbricoides* **(%)** | **0** | **2.6** | | | |
| *Trichuris trichiura* **(%)** | **0** | **10.3** | | | |
| *Necator americanus* **(%)** | **0** | **7.7** | | | |
| *Strongyloides* **(%)** | **0** | **2.6** | | | |
| **TIgE (IU/L)** (median, IQR)* | **6216** (1545–17478) | **10476** (6570–19740) | **0.034** | 3226 (-1584, 8035) | 0.19 |
| **Eosinophils (%)** (mean, SD)* | **10.3** (8.2) | **18.0** (9.6) | **0.011** | **7.1** (0.9, 13.4) | **0.025** |
| **hs-CRP (mg/L)** (median, IQR) | **1.91** (0.49–4.52) | **1.71** (0.68–4.07) | 0.78 | **-0.12** (-0.81, 0.56) | 0.72 |
| **ALAT (GPT, U/L)** (mean, SD) | **19.2** (11.9) | **17.1** (9.3) | 0.42 | **-3.3** (-8.4, 1.8) | 0.20 |
| **ASAT (GOT, U/L)** (mean, SD) | **26.4** (10.1) | **24.2** (7.0) | 0.29 | **-2.9** (-7.0, 1.2) | 0.16 |
| **Glucose (mmol/L)** (mean, SD) | **4.61** (1.06) | **4.52** (0.67) | 0.66 | **-0.14** (-0.57, 0.28) | 0.51 |
| **Insulin (mU/L)** (median, IQR) | **4.45** (2.74–7.21) | **4.54** (2.91–9.62) | 0.91 | **0.06** (-0.43, 0.55) | 0.81 |
| **C-peptide (nmol/L)** (median, IQR) | **0.41** (0.32–0.61) | **0.36** (0.27–0.62) | 0.59 | **-0.05** (-0.38, 0.27) | 0.75 |
| **HOMA-IR** (median, IQR) | **0.94** (0.52–1.32) | **1.00** (0.57–1.80) | 0.87 | **0.06** (-0.45, 0.58) | 0.80 |
| **TC (mmol/L)** (mean, SD) | **4.42** (0.84) | **4.01** (0.81) | **0.037** | **-0.30** (-0.68, -0.08) | 0.11 |
| **HDL-C (mmol/L)** (mean, SD) | **1.44** (0.40) | **1.18** (0.31) | **0.003** | **-0.24** (-0.43, -0.06) | **0.009** |
| **LDL-C (mmol/L)** (mean, SD) | **2.56** (0.73) | **2.50** (0.78) | 0.74 | **-0.04** (-0.31, 0.38) | 0.84 |
| **TG (mmol/L)** (mean, SD) | **0.93** (0.50) | **0.72** (0.21) | **0.031** | **-0.20** (-0.39, -0.03) | **0.022** |

Normally distributed data are presented as means +/- standard deviation (SD) and non-normally distributed data as median +/- interquartile range (IQR). Adjusted mean difference for TIgE, hs-CRP, Insulin, C-peptide, and HOMA-IR were anti-log transformed.

*, some values are missing (n = 31 in *Sh-* for TIgE; n = 18 in *Sh-* and n = 30 in *Sh+* for eosinophils). Abbreviations: BMI: body mass index, TIgE: total immunoglobulin E, hs-CRP: high-sensitivity C-reactive protein, ALAT: alanine aminotransferase, ASAT: aspartate aminotransferase, HOMA-IR: HOmeostatic Model Assessment for Insulin Resistance, TC: total cholesterol, HDL-C: high density lipoprotein-cholesterol, LDL-C: low density lipoprotein cholesterol, TG: triglycerides.

(+54%;*P* = 0.025) and blood eosinophils (+57%;*P* = 0.028) was observed in *S. haematobium*-infected individuals (*Sh*⁺).

## Effects of *S. haematobium* infection on HOMA-IR, circulating cholesterol and triglyceride levels

To assess the impact of *S. haematobium* infection on glucose homeostasis and insulin resistance, the serum levels of glucose, insulin and C-peptide were measured in *Sh*⁻ and *Sh*⁺ individuals and the HOMA-IR index calculated. As shown in Table 1, these parameters were not different between the two groups. Next, the effect of *S. haematobium* infection on lipid homeostasis was assessed and serum TC (-9%;*P* = 0.037), HDL-C (-18%;*P* = 0.003) and TG (-23%;*P* = 0.031) levels were found to be lower in *Sh*⁺ when compared to *Sh*⁻ individuals. After adjustment for age, sex, BMI, and other helminth infection, we observed similar effects, although it remained only significant for HDL-C and TG (Table 1). Multivariate logistic regression analysis was also performed to assess the influence of S. haematobium infection on the abovementioned metabolic parameters and gave similar outcomes (S1 Table). Importantly, the reductions in serum TC, HDL-C and TG levels were mostly seen in overweight/obese infected individuals when the population was stratified according to low/normal (<25) or high (>25) BMI (Table 2). Of note, removing the 7 Sh⁺ individuals with other helminth infection from the analyses has only marginal impact on the results and does not affect our conclusions.

## Association between *S. haematobium* infection intensity and serum lipid levels

When measuring serum CAA concentrations, a highly sensitive method for assessing infection intensity [20], we found that a significant fraction of *Sh*⁻ individuals (34%) scored positive for CAA (>10pg/ml), indicating active worm infection, whereas few *Sh*⁺ individuals (9%) returned a CAA concentration below the threshold value of the assay (S2 Fig).

After re-analysis of the data using the serum CAA levels as diagnostic criteria, we found that *S. haematobium* infection was still associated with significant increase in blood eosinophils (+70%;*P* = 0.040) and reduction in serum TC (-10%;*P* = 0.020) levels after adjustment for age, sex and BMI (Table 3). Multivariate logistic regression analysis gave similar outcomes (S2 Table). Similar outcomes were found when stratifying the population according to the intensity of the infection using CAA levels, with some non-significant trends for reduced serum levels also observed for HDL-C, LDL-C and TG (S3 Table). When stratifying the population according to BMI, a significant decrease in serum TC levels was observed only in overweight/obese infected individuals (S4 Table). In line with this, significant negative correlations were found between serum CAA and TC, HDL-C, LDL-C and TG levels, respectively, in both non-stratified whole population and obese/overweight individuals but not in lean subjects (Fig 1).

In order to further investigate the effects of *S. haematobium* infection on circulating lipids and lipoproteins, we performed comprehensive quantitative lipidomic analysis of serum from lean and overweight/obese individuals. The levels of the individual lipid species (S5 Table) belonging to the same lipid class were summed up for an approximate measure of the total serum levels for each lipid class (Fig 2A–2B). We confirmed that total cholesteryl ester (CE) and TG levels were significantly higher in overweight/obese individuals when compared to lean subjects. A significant reduction of CE was observed in both groups of infected subjects but only correlated with the intensity of infection in obese/overweight individuals. Total phosphatidylcholine (PC) and TG levels were also found to be reduced in obese/overweight but not lean infected individuals (Fig 2A–2B). When zooming into the individual lipid species, *S. haematobium* infection was seen to be associated with a significant reduction of 6 PCs (Fig 2C), 3

**Table 2. Characteristics of the study population stratified according to body mass index.**

| | BMI <25 | | | | | BMI >25 | | | | |
|---|---|---|---|---|---|---|---|---|---|---|
| | *S. haematobium* negative (n = 17) | *S. haematobium* positive (n = 22) | P-value | *Mean difference adjusted for age, sex and other helminths (95% CI)* | P-value | *S. haematobium* negative (n = 15) | *S. haematobium* positive (n = 17) | P-value | *Mean difference adjusted for age, sex and other helminths (95% CI)* | P-value |
| **Age (year)** (mean, range) | **34.3** (18–63) | **34.5** (18–58) | 0.96 | | | **37.4** (19–49) | **34.5** (18–63) | 0.45 | | |
| **Male (%)** | **41.2** | **50.0** | | | | **46.7** | **47.1** | | | |
| **BMI (kg/m²)** (mean, SD) | **21.6** (2.4) | **22.2** (1.8) | 0.40 | **0.8** (-0.7, 2.2) | 0.30 | **32.6** (5.6) | **30.0** (3.0) | 0.11 | **-2.8** (-5.8, 0.2) | 0.06 |
| ***S. haematobium* urine eggs** (median, IQR) | **0** | **10** (4–62) | <0.001 | | | **0** | **30** (5–59) | <0.001 | | |
| **Other helminths (%)** | **0** | **13.6** | | | | **0** | **23.5** | | | |
| *Ascaris lumbricoides* (%) | 0 | 0 | | | | 0 | 5.9 | | | |
| *Trichuris trichiura* (%) | 0 | 4.5 | | | | 0 | 17.6 | | | |
| *Necator americanus* (%) | 0 | 9.1 | | | | 0 | 5.9 | | | |
| *Strongyloides* (%) | 0 | 4.5 | | | | 0 | 0 | | | |
| **TIgE (IU/L)** (median, IQR)* | **5912** (1446–19547) | **10470** (5589–21261) | 0.16 | **2156** (-4263, 8576) | 0.50 | **6562** (1447–11527) | **10476** (6899–18669) | 0.07 | **3916** (-4020, 11852) | 0.32 |
| **Eosinophils (%)** (mean, SD)* | **12.0** (9.0) | **20.3** (9.8) | 0.08 | **8.3** (-2.0, 18.6) | 0.11 | **9.2** (8.0) | **15.1** (8.7) | 0.12 | **5.6** (-3.1. 14.2) | 0.19 |
| **hs-CRP (mg/L)** (median, IQR) | **2.15** (0.55–6.82) | **1.19** (0.52–3.42) | 0.28 | **-0.52** (-1.50, 0.46) | 0.29 | **1.79** (0.46–3.78) | **2.63** (1.06–5.96) | 0.28 | **0.31** (-0.62, 1.24) | 0.50 |
| **ALAT (GPT, U/L)** (mean, SD) | **17.0** (10.6) | **16.2** (5.9) | 0.76 | **-1.1** (-6.8., 4.6) | 0.69 | **21.6** (13.2) | **18.3** (12.5) | 0.47 | **-6.2** (-15.8.9, 3.3) | 0.19 |
| **ASAT (GOT, U/L)** (mean, SD) | **25.6** (10.5) | **23.9** (6.1) | 0.53 | **-2.5** (-8.0, 3.0) | 0.36 | **27.3** (10.0) | **24.7** (8.2) | 0.42 | **-3.5** (-10.3, 3.2) | 0.29 |
| **Glucose (mmol/L)** (mean, SD) | **4.47** (1.19) | **4.60** (0.62) | 0.66 | **0.10** (-0.50, 0.70) | 0.73 | **4.76** (0.91) | **4.41** (0.73) | 0.23 | **-0.49** (-1.14, 0.15) | 0.13 |
| **Insulin (mU/L)** (median, IQR) | **4.55** (2.78–7.93) | **5.08** (2.91–9.88) | 0.77 | **0.06** (-0.59, 0.71) | 0.84 | **4.22** (2.68–7.16) | **3.76** (2.44–8.55) | 0.87 | **0.06** (-0.75, 0.87) | 0.88 |
| **C-peptide (nmol/L)** (median, IQR) | **0.38** (0.31–0.58) | **0.43** (0.29–0.68) | 0.93 | **0.02** (-0.44, 0.49) | 0.92 | **0.42** (0.32–1.07) | **0.32** (0.26–0.54) | 0.36 | **-0.12** (-0.63, 0.39) | 0.63 |
| **HOMA-IR** (median, IQR) | **0.93** (0.49–1.33) | **1.10** (0.59–2.01) | 0.58 | **0.15** (-0.54, 0.83) | 0.66 | **0.99** (0.52–1.32) | **0.80** (0.44–1.71) | 0.73 | **-0.04** (-0.89, 0.81) | 0.92 |
| **TC (mmol/L)** (mean, SD) | **4.11** (0.75) | **3.98** (0.70) | 0.60 | **-0.06** (-0.49, 0.36) | 0.76 | **4.78** (0.81) | **4.04** (0.96) | **0.025** | **-0.52** (-1.16, 0.12) | 0.11 |
| **HDL-C (mmol/L)** (mean, SD) | **1.43** (0.47) | **1.21** (0.36) | 0.12 | **-0.23** (-0.52, 0.07) | 0.12 | **1.46** (0.32) | **1.13** (0.22) | **0.002** | **-0.27** (-0.48, -0.06) | **0.013** |
| **LDL-C (mmol/L)** (mean, SD) | **2.30** (0.54) | **2.43** (0.69) | 0.52 | **0.21** (-0.16, 0.57) | 0.26 | **2.85** (0.82) | **2.59** (0.89) | 0.39 | **-0.10** (-0.74, 0.53) | 0.74 |
| **TG (mmol/L)** (mean, SD) | **0.83** (0.24) | **0.74** (0.23) | 0.22 | **-0.10** (-0.25, 0.05) | 0.18 | **1.05** (0.69) | **0.70** (0.18) | 0.07 | **-0.31** (-0.66, 0.05) | 0.08 |

Normally distributed data are presented as means +/- standard deviation (SD) and non-normally distributed data as median +/- interquartile range (IQR). Adjusted mean difference for TIgE, hs-CRP, Insulin, C-peptide and HOMA-IR were anti-log transformed.

*, some values are missing (for TIgE n = 14 in BMI>25 and *Sh*-; for eosinophils n = 6 in BMI<25 and *Sh*-, n = 17 in BMI<25 and *Sh*+ and n = 9 in BMI>25 and *Sh*-).

Abbreviations: BMI: body mass index; TIgE: total immunoglobulin E; hs-CRP: high-sensitivity C-reactive protein; ALAT: alanine aminotransferase; ASAT: aspartate aminotransferase; HOMA-IR: HOmeostatic Model Assessment for Insulin Resistance; TC: total cholesterol; HDL-C: high density lipoprotein-cholesterol; LDL-C: low density lipoprotein cholesterol; TG: triglycerides.

**Table 3. Characteristics of the study population stratified according to CAA levels.**

| | CAA<10 pg/ml (n = 21) | CAA>10 pg/ml (n = 50) | *P*-value | Mean difference adjusted for age, sex and BMI (95% CI) | *P*-value |
|---|---|---|---|---|---|
| **Age (year)** (mean, range) | **39.2** (18–63) | **33.3** (18–63) | 0.07 | | |
| **Male (%)** | **42.9** | **48** | | | |
| **BMI (kg/m²)** (mean, SD) | **27.0** (6.8) | **25.7** (5.2) | 0.40 | | |
| **TIgE (IU/L)** (median, IQR)* | **5559** (700–17761) | **10207** (6104–18408) | **0.025** | **4485** (-618, 9588) | 0.08 |
| **Eosinophils (%)** (mean, SD)* | **9.9** (7.5) | **17.3** (9.8) | **0.028** | **6.9** (0.3, 13.4) | **0.040** |
| **hs-CRP (mg/L)** (median, IQR) | **1.66** (0.50–2.69) | **2.27** (0.68–5.07) | 0.24 | **0.58** (-0.13, 1.29) | 0.11 |
| **ALAT (GPT, U/L)** (mean, SD) | **16.3** (7.3) | **18.8** (11.7) | 0.38 | **3.3** (-1.6, 8.2) | 0.18 |
| **ASAT (GOT, U/L)** (mean, SD) | **23.0** (5.3) | **26.2** (9.5) | 0.08 | **2.43** (-1.9, 6.8) | 0.20 |
| **Glucose (mmol/L)** (mean, SD) | **4.44** (1.12) | **4.61** (0.73) | 0.44 | **0.30** (-0.15, 0.75) | 0.18 |
| **Insulin (mU/L)** (median, IQR) | **4.55** (2.76–7.19) | **4.44** (2.89–9.63) | 0.94 | **0.09** (-0.43, 0.61) | 0.73 |
| **C-peptide (nmol/L)** (median, IQR) | **0.41** (0.31–0.69) | **0.37** (0.27–0.62) | 0.33 | **-0.12** (-0.47, 0.22) | 0.48 |
| **HOMA-IR** (median, IQR) | **0.94** (0.46–1.22) | **0.95** (0.57–1.80) | 0.71 | **-0.19** (-0.68, 0.29) | 0.43 |
| **TC (mmol/L)** (mean, SD) | **4.62** (0.90) | **4.01** (0.80) | **0.005** | **-0.47** (-0.86, -0.07) | **0.020** |
| **HDL-C (mmol/L)** (mean, SD) | **1.42** (0.38) | **1.24** (0.36) | 0.06 | **-0.18** (-0.38, 0.02) | 0.07 |
| **LDL-C (mmol/L)** (mean, SD) | **2.78** (0.82) | **2.42** (0.70) | 0.06 | **-0.24** (-0.61, 0.12) | 0.19 |
| **TG (mmol/L)** (mean, SD) | **0.91** (0.56) | **0.78** (0.28) | 0.31 | **-0.09** (-0.28, 0.10) | 0.34 |

Normally distributed data are presented as means +/- standard deviation (SD) and non-normally distributed data as median +/- interquartile range (IQR). Adjusted mean difference for TIgE, hs-CRP, Insulin, C-peptide and HOMA-IR were anti-log transformed.

*, some values are missing (n = 49 in CAA>10pg/ml for TIgE; n = 11 in CAA<10pg/ml and n = 34 in CAA>10pg/ml for eosinophils). Abbreviations: BMI: body mass index; TIgE: total immunoglobulin E; CAA: circulating anodic antigen; hs-CRP: high-sensitivity C-reactive protein; ALAT: alanine aminotransferase; ASAT: aspartate aminotransferase; HOMA-IR: HOmeostatic Model Assessment for Insulin Resistance; TC: total cholesterol; HDL-C: high density lipoprotein-cholesterol; LDL-C: low density lipoprotein cholesterol; TG: triglycerides.

CEs (Fig 2D) and 6 TGs (Fig 2E), especially among the most abundant species which are also found to be increased in overweight/obese individuals (*e.g.* PC(16:0/18:1), CE (18:2) and TG (52:2))[21, 22].

## Discussion

The main objective of the present study was to investigate the effects of *S. haematobium* infection on metabolic outcomes, especially on whole-body lipid/lipoprotein homeostasis, in lean and overweight/obese individuals residing in a highly endemic rural area. Although no effect of *S. haematobium* infection was observed on HOMA-IR, whatever the BMI status, a strong reduction in serum TC levels was consistently observed in infected individuals. Furthermore, negative correlations were also observed between the intensity of active *S. haematobium* infection and the circulating levels of TC, HDL-c LDL-c, TG, and some lipid species associated with cholesterol-rich lipoprotein particles in overweight/obese individuals. Altogether, our findings showed for the first time that *S. haematobium* infection is associated with improved circulating lipid profiles in humans, especially in overweight/obese individuals.

The lowering effect of *S. haematobium* infection on serum TC and cholesterol levels observed in our study is in line with the few cross-sectional investigations conducted so far in countries endemic for other helminth species. Wiria *et al.* have indeed shown that infection with the soil-transmitted helminths *Necator americanus*, *Ascaris lumbricoides* and *Trichuris trichiura* was associated with reduced serum levels of TC and HDL-c in lean Indonesian individuals from Flores Island [23]. Duan *et al.* have also recently reported that serum TG, TC and

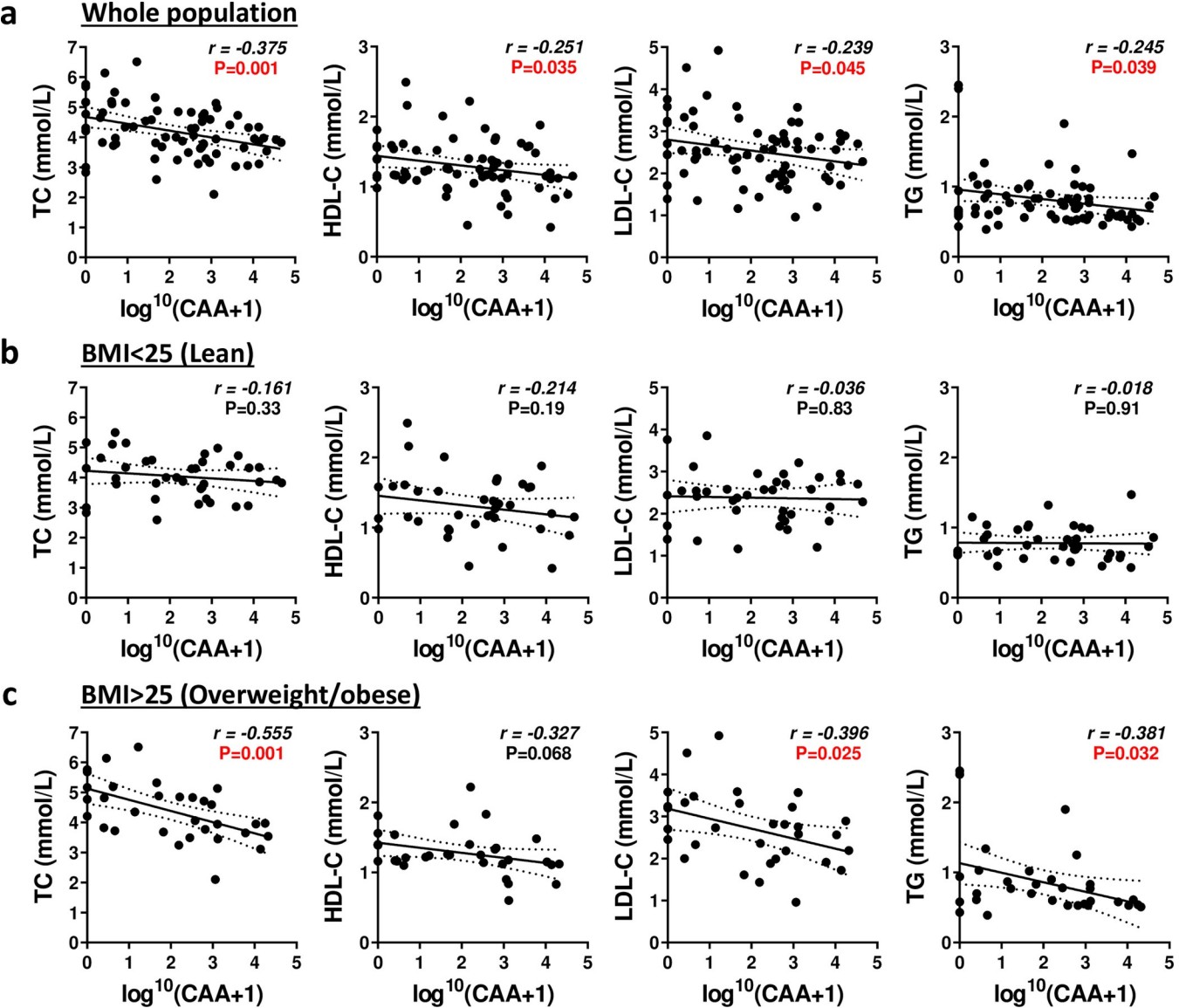

**Fig 1. Associations between intensity of *S. haematobium* infection assessed by circulating anodic antigen levels and serum lipid parameters.** The correlations between serum CAA and TC, HDL-C, LDL-C and TG were shown for the whole population ([n = 71], a) and for lean (BMI<25 [n = 39], b) and overweight/obese (BMI>25 [n = 32], c) individuals. *Sh*, *Schistosoma haematobium;* CAA, Circulating Anodic Antigen; TC, Total Cholesterol; HDL-C, High-Density Lipoprotein-Cholesterol; LDL-C, Low-Density Lipoprotein-Cholesterol; TG, Triglycerides.

LDL-c levels were significantly lower in lean Chinese subjects with chronic *S. japonicum* infection when compared to uninfected individuals [24]. Finally, during the preparation of the manuscript, the results of a study investigating the effects of *S. mansoni* infection on metabolic outcomes in the framework of the LaVIISWA trial conducted in the highly-endemic region of Lake Victoria in Uganda became available [25]. Interestingly, a negative association between the intensity of *S. mansoni* infection and serum TC, HDL-c, LDL-c and TG levels was found [25]. Moreover, community-wide intensive anthelminthic treatment that resulted in lower *S. mansoni* prevalence was also associated with a trend towards increased TC and LDL-c, suggesting that deworming might revert some of the effects induced by helminth infection on lipid profile [25]. However, it is worth to mentioning that our study is the first one

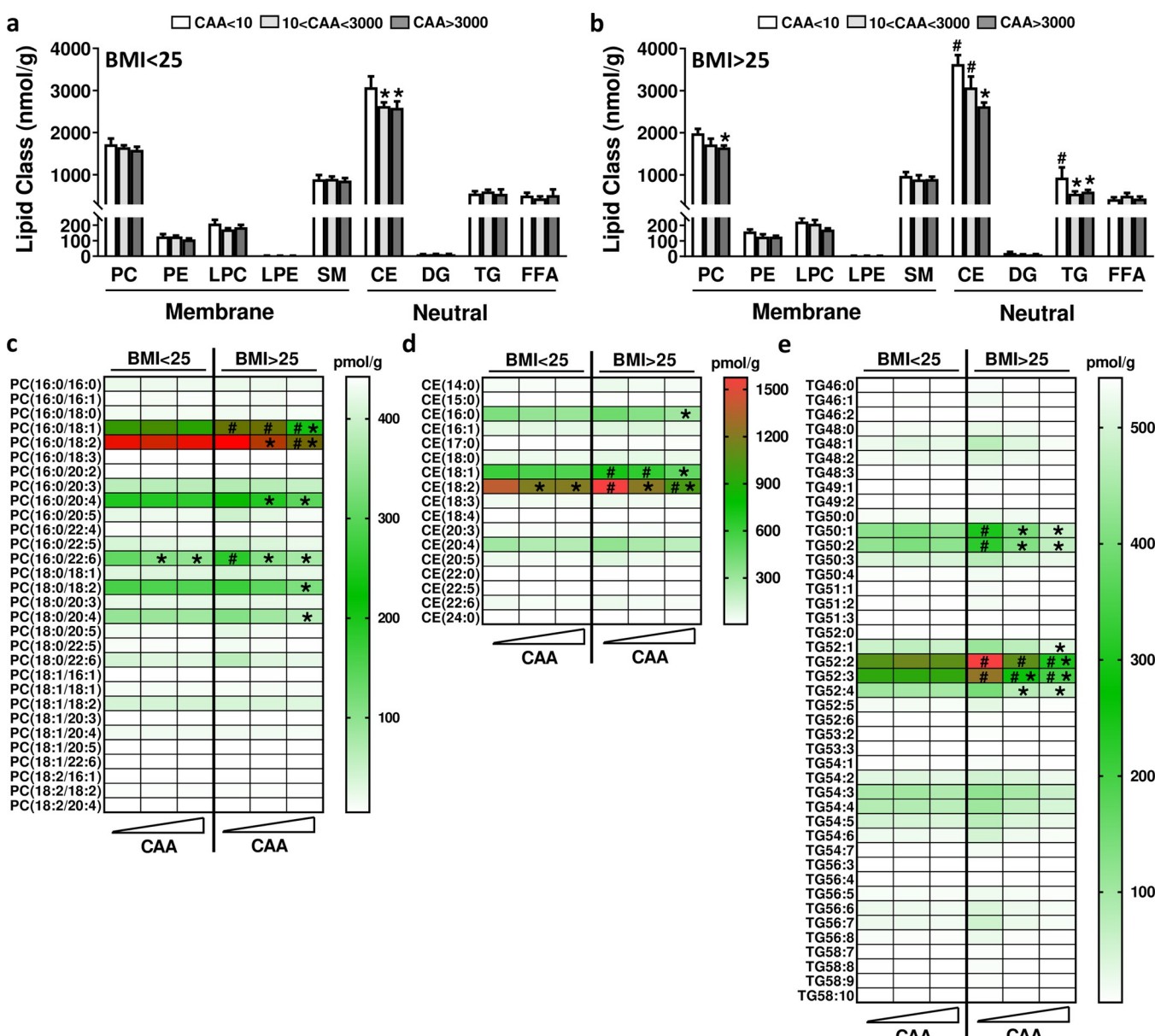

**Fig 2. Effects of *S. haematobium* infection on serum lipid profiles in lean and obese subjects.** Lipidomics analysis was performed using the Lipidyzer platform on serum from lean (BMI<25, a) and overweight/obese (BMI>25, b) individuals, and the data for the lipid classes (a) and for the various PC (c), CE (d) and TG (e) species were stratified in each population according to 'below detection threshold' (<10pg/ml), 'medium/high' (10<CAA<3000pg/ml) or 'high' (>3000pg/ml) CAA levels. PC, Phosphatidylcholine; PE, Phosphatidylethanolamine; LPC, Lysophosphatidylcholine; LPE, Lysophosphatidylethanolamine; SM, Sphingomyelin; CE, Cholesterylester; DG, Diglycerides; TG, Triglycerides; FFA, Free-fatty acids. *, p<0.05 *versus* CAA < 10; #, p<0.05 *versus* BMI <25.

investigating the impact of helminth infection in overweight/obese individuals, who have elevated lipid/cholesterol levels and higher risk for developing cardiometabolic diseases.

Several underlying mechanisms may account for the effects of helminth on lipid homeostasis, involving 1) parasitism of host dietary nutrients through hijacking of lipid resources for their own survival and reproduction, 2) alteration of gut microbiota and intestinal lipid metabolism, and 3) immune-dependent or -independent regulation of lipid metabolism in immune cells and/or peripheral tissues by helminths and their molecules.

Although adult schistosomes cannot oxidize exogenous FA [26], they can use and modify host lipid for biosynthetic purposes by scavenging cholesterol and incorporating FA as phospholipids and neutral lipids in the worm eggs, notably for membrane formation of the developing miracidium [27]. It is therefore tempting to speculate that part of the reduction in some lipoprotein and TG species observed in *S. haematobium*-infected individuals might result from their active transfer to the eggs produced by female worms and their subsequent elimination in the urine. Further studies are however required to investigate this point.

During the last decade, the gut microbiota has emerged as an important player in host metabolic homeostasis, notably by regulating intestinal lipid processing from dietary fat and nutrient metabolism in peripheral organs, but also in shaping the local and systemic immune response [28, 29]. Gut microbiota, via local production of various metabolites, can impact host lipid metabolism by modulating bile acid metabolism and/or a gut-brain axis involved in the control of whole-body lipid homeostasis [30, 31]. Since various helminth species, including *S. haematobium* [32], have been reported to affect gut microbiota in humans [33], one may hypothesize that this could underlie some of the reduction in TC and TG levels observed in helminth infected individuals, secondary to alterations of intestinal lipid absorption, hepatic cholesterol and BA biosynthesis, secretion and reuptake, or hydrolysis of TG-rich lipoproteins by peripheral metabolic tissues. Further studies would be required for investigating the putative associations between helminth-induced changes in specific gut bacterial species and serum/fecal lipid/BA profiles.

Infection with schistosomes is associated with eosinophilia, parasite-specific IgE and elevated type-2 cytokines, which are all hallmarks of a $T_H2$ immune response [10, 34]. One may speculate that some of the beneficial effects observed on serum lipid profile from obese/overweight individuals could result from this helminth-induced immunomodulation through direct or indirect on metabolic organs. The liver plays an essential role in lipid/cholesterol metabolism, notably by regulating *de novo* lipogenesis, cholesterol biosynthesis and VLDL-TG secretion, and facilitating the clearance of circulating HDL-c via selective uptake through membrane scavenger receptors. Interestingly, type-2 cytokines, such as IL-4 and IL-13, have been shown to regulate hepatic metabolism in rodents through direct interaction with their cognate receptors expressed on hepatocytes [35, 36]. This type-2 micro-environment could also have some impact on lipid metabolism in other peripheral organs, notably in AT through modulation of TG lipolysis/lipogenesis. It is also worth mentioning that some of the *S. haematobium* worm- and/or egg-derived molecules released inside the host may also contribute to tissue-specific modulation of lipid metabolism through immune-independent direct interaction with metabolic cells [11, 37]. Furthermore, intracellular nutrient metabolism has been shown to be central in the regulation of immune cell activation, differentiation and function [38, 39]. As such, one may speculate that the decrease in serum lipid levels observed in *S. haematobium*-infected individuals may also partly result from increased FA and cholesterol metabolism by various circulating and/or tissue-resident immune cells secondary to helminth-induced host immunomodulation, as reported for example in hepatic macrophages from infected mice [40].

The main limitations of our study are based on its observational nature, which prevents firm conclusions on causal relationship, and the relatively small sample size, especially after stratification according to BMI or CAA levels. Future studies involving larger cohort of lean and obese individuals and/or deworming intervention using anti-helminthic drug in such population would be required. In contrast to what has been reported with STH [12, 13, 16], we did not find any association between *S. haematobium* infection and improved whole-body insulin resistance as assessed by HOMA-IR. This discrepancy could be explained by differences in helminth species but also eventually in age and/or lifestyle of the population studied,

notably their nutritional habits. Furthermore, past infections with *S. haematobium* could have already imprinted their effect on insulin sensitivity, masking the influence of current infection. In this regard, trials where anthelminthic treatment is used or controlled human infection in naïve individuals with various helminth species [41–43] can better answer this question.

## Supporting information

**S1 Fig. Flow chart of study participants enrolment.** Sh, *Schistosoma haematobium*; STH, Soil-transmitted helminths; BMI, Body Mass Index.
(TIF)

**S2 Fig. Serum CAA levels in urine egg negative and positive individuals.** The serum CAA concentrations were measured in Sh- and Sh+ individuals diagnosed by presence of urine eggs (a), and the data were stratified in each population according to 'below detection threshold' (<10pg/ml), 'medium/high' (10<CAA<3000pg/ml) or 'high' (>3000pg/ml) CAA levels (b). The correlation between serum CAA levels and urine *S. haematobium* egg counts (c) in the whole population was plotted.
(TIF)

**S1 Table. Multiple logistic regression analysis on effect of *Schistosoma haematobium* infection diagnosed by urine egg microscopy on various metabolic parameters.** Abbreviations: CI: confidence interval; OR: Odd ratio; TIgE: total immunoglobulin E; hs-CRP: high-sensitivity C-reactive protein; ALAT: alanine aminotransferase; ASAT: aspartate aminotransferase; HOMA-IR: HOmeostatic Model Assessment for Insulin Resistance; TC: total cholesterol; HDL-C: high density lipoprotein-cholesterol; LDL-C: low density lipoprotein cholesterol; TG: triglyceride.
(DOCX)

**S2 Table. Multiple logistic regression analysis on effect of *Schistosoma haematobium* infection diagnosed by serum CAA levels on various metabolic parameters.** Abbreviations: CI: confidence interval; OR: Odd ratio; TIgE: total immunoglobulin E; hs-CRP: high-sensitivity C-reactive protein; ALAT: alanine aminotransferase; ASAT: aspartate aminotransferase; HOMA-IR: HOmeostatic Model Assessment for Insulin Resistance; TC: total cholesterol; HDL-C: high density lipoprotein-cholesterol; LDL-C: low density lipoprotein cholesterol; TG: triglyceride.
(DOCX)

**S3 Table. Characteristics of the study population stratified according to CAA range.** Normally distributed data are presented as means +/- standard deviation (SD) and non-normally distributed data as median +/- interquartile range (IQR). *, some values are missing (for TIgE n = 36 in 10<CAA<3000pg/ml; for eosinophils n = 11 in CAA<10pg/ml, n = 24 in 10<CAA<3000pg/ml and n = 10 in CAA>3000pg/ml). Abbreviations: BMI: body mass index; TIgE: total immunoglobulin E; CAA: circulating anodic antigen; hs-CRP: high-sensitivity C-reactive protein; ALAT: alanine aminotransferase; ASAT: aspartate aminotransferase; HOMA-IR: HOmeostatic Model Assessment for Insulin Resistance; TC: total cholesterol; HDL-C: high density lipoprotein-cholesterol; LDL-C: low density lipoprotein cholesterol; TG: triglyceride
(DOCX)

**S4 Table. Characteristics of the study population stratified according to CAA levels and body mass index.** Normally distributed data are presented as means +/- standard deviation (SD) and non-normally distributed data as median +/- interquartile range (IQR). Adjusted

mean difference for TIgE, hs-CRP, Insulin, C-peptide and HOMA-IR were anti-log transformed. *, some values are missing (for TIgE n = 21 in BMI>25 and CAA>10pg/ml, for eosinophils n = 4 in BMI<25 and CAA<10pg/ml, n = 19 in BMI<25 and CAA>10pg/ml, n = 7 in BMI>25 and CAA<10pg/ml and n = 15 in BMI>25 and CAA>10pg/ml). Abbreviations: BMI: body mass index; TIgE: total immunoglobulin E; CAA: circulating anodic antigen; hs-CRP: high-sensitivity C-reactive protein; ALAT: alanine aminotransferase; ASAT: aspartate aminotransferase; HOMA-IR: HOmeostatic Model Assessment for Insulin Resistance; TC: total cholesterol; HDL-C: high density lipoprotein-cholesterol; LDL-C: low density lipoprotein cholesterol; TG: triglycerides.
(DOCX)

**S5 Table. Serum lipidomics in subjects stratified according to CAA levels.** Data are presented as means (+/- SD). Abbreviations: CAA: circulating anodic antigen; PC: Phosphatidylcholine; PE: Phosphatidylethanolamine; LPC: Lysophosphatidylcholine; LPE: Lysophosphatidylethanolamine; SM: Sphingomyelin; CE: Cholesterylester; DG: Diglycerides; TG: Triglycerides; FFA: Free-fatty acids.
(DOCX)

## Acknowledgments

The authors want to thank the participants who took part in this study.

## Author Contributions

**Conceptualization:** Bruno Guigas.

**Data curation:** Jeannot F. Zinsou, Bruno Guigas.

**Formal analysis:** Jeannot F. Zinsou, Jacqueline J. Janse, Bruno Guigas.

**Funding acquisition:** Martin Giera, Maria Yazdanbakhsh, Bruno Guigas.

**Investigation:** Jeannot F. Zinsou, Yabo Y. Honpkehedji, Jean Claude Dejon-Agobé, Noemí García-Tardón, Pytsje T. Hoekstra, Marguerite Massinga-Loembe, Martin Giera.

**Methodology:** Jeannot F. Zinsou, Noemí García-Tardón, Pytsje T. Hoekstra, Paul L. A. M. Corstjens, Govert J. van Dam, Martin Giera, Bruno Guigas.

**Project administration:** Jeannot F. Zinsou, Marguerite Massinga-Loembe, Ayola A. Adegnika, Bruno Guigas.

**Supervision:** Peter G. Kremsner, Maria Yazdanbakhsh, Ayola A. Adegnika, Bruno Guigas.

**Visualization:** Bruno Guigas.

**Writing – original draft:** Jeannot F. Zinsou, Bruno Guigas.

**Writing – review & editing:** Jean Claude Dejon-Agobé, Pytsje T. Hoekstra, Paul L. A. M. Corstjens, Martin Giera, Maria Yazdanbakhsh, Ayola A. Adegnika, Bruno Guigas.

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
