## [Decision Letter · Decision Letter 0]

29 Apr 2020

Dear Dr. Guigas,

Thank you very much for submitting your manuscript "Schistosoma haematobium infection is associated with lower serum cholesterol levels and improved lipid profile in overweight/obese individuals" for consideration at PLOS Neglected Tropical Diseases. As with all papers reviewed by the journal, your manuscript was reviewed by members of the editorial board and by several independent reviewers. The reviewers appreciated the attention to an important topic. Based on the reviews, we are likely to accept this manuscript for publication, providing that you modify the manuscript according to the review recommendations. 

This study provides insight into the timely topic of helminth induced modulation of host metabolism. The reviewers are overall enthusiastic about the manuscript, but have a few minor issues that need to be addressed for acceptance. Please address the points raised by Reviewers 2 and 3 on sample size, co-infections, the use of CAA, and expanding the discussion to align the current study with the recent literature. As well as addressing the mislabeling raised by reviewers 1 and 3.

Sincerely,

Keke Fairfax, PhD

Guest Editor

Michael Hsieh

Deputy Editor

This study provides insight into the timely topic of helminth induced modulation of host metabolism. The reviewers are overall enthusiastic about the manuscript, but have a few minor issues that need to be addressed for acceptance. Please address the points raised by Reviewers 2 and 3 on sample size, co-infections, the use of CAA, and expanding the discussion to align the current study with the recent literature. As well as addressing the mislabeling raised by reviewers 1 and 3.

Reviewer's Responses to Questions

**Key Review Criteria Required for Acceptance?**

**Methods**

-Are the objectives of the study clearly articulated with a clear testable hypothesis stated?

-Is the study design appropriate to address the stated objectives?

-Is the population clearly described and appropriate for the hypothesis being tested?

-Is the sample size sufficient to ensure adequate power to address the hypothesis being tested?

-Were correct statistical analysis used to support conclusions?

-Are there concerns about ethical or regulatory requirements being met?

Reviewer #1: Methods are appropriate and the study design addresses the stated objectives. Although the population size is small

Reviewer #2: The authors have performed a well designed, reasonably well powered and thorough analysis of metabolic parameters in each cohort. The statistical analysis is appropriate and of excellent standard. The study is novel in regards to the depth of analysis, and the human population (ie concurrent analysis of obese and lean individuals +/- worms.

One confounder is that is addition to S. hematobium (Sh), 7 of the participants also had another helminth infection. This is acknowledged by the authors, but it means that within their Sh+ cohort, there are people with Sh only, and some people with Sh plus other parasites. This may have implications for their results, and the authors should present some data whether the presence of a con-infection leads to quantitative differences in any of the key parameters.

Reviewer #3: Study population: 

The authors do not mention how the sample size was calculated. Please include this information in the Methods section. One of the main limitations of this study is the sample size. This information is crucial to determine if the sample size is appropriate, and therefore the conclusions can be supported. 

Individuals Sh+ were negative for STH and Plasmodium? It is not clear in the Methods if co-infections were excluded. Line 112: for treatment of Sh+ individuals, parasitological (presence of eggs in urine) and CAA results were considered?

Hematological and biochemical analysis: 

I strongly recommend including the reference values of the biochemical parameters for your study population as supplemental material. 

Quantitative insulin sensitivity check index (QUICKI) = 1 / (log(fasting insulin μU/mL) + log(fasting glucose mg/dL)) should be calculated and included as an additional measurement of insulin resistance.

**Results**

-Does the analysis presented match the analysis plan?

-Are the results clearly and completely presented?

-Are the figures (Tables, Images) of sufficient quality for clarity?

Reviewer #1: Results are clearly and completely presented. One aspect not addressed is the relationship of the immune response (as determined by eosinophils numbers and or %) to the lipid profile for each individual. The authors do mention in the discussion that the possible effect of IL4/IL13 on hepatocyte function may be one of the mechanisms for lowering TG. Eosinophil levels do reflect the immune response to the parasite by an individual and although the authors are correct to use CAA levels to measure intensity of parasite infection it would also be worth it I thought to measure intensity of the immune response to the Sh and lipid levels.

The 'Statistical Analysis' paragraph is well written. These are a good choice of tests, consideration of confounding factors, and multiple testing corrections. 

Tables S1 & S2 is where they show the raw Odds Ratio and confounder adjusted OR for egg and serum levels. Tables S1, S2 are very informative, and are well presented by showing both raw and adjusted results (that consider multiple important confounding factors). 

Tables S3, S4, S5 are mislabelled. Table S3 summarises factors stratified by CAA range; the Eosinophil response appears to show incredibly strong correlation to stratification level, it is unfortunate they lost some measurements for Eosinophils as stated, but their choice of statistical test is well suited to different sample sizes. Table S4 shows the disparity between gender between BMI >/< 25 groups, showing the importance of showing adjusted OR – It is important that the presentation of raw data is included.

Fig. 1 It is interesting when whole population the HDL-C has a significant p-value but when you split it into lean & obese there is no significant p-value in either, despite the same trend. The overall trend is clear however that serum CAA levels are associated with many cholesterol measurements in the obese category. 

Fig. 2 a and b are skillfully plotted - and a statement is needed with respect to what p-values are associated to (#/*/#*) on the heatmap.

Reviewer #2: The results are well presented and clear

Reviewer #3: The CAA corrected data should be included in the main paper and not as supplemental materials. The authors should simplify the results and use only the CAA positive and CAA negative as the Sh+ and Sh- groups in the paper. Table 2 should be corrected using the CAA data. 

Figure 1: Please include the number of patients for each analysis (whole population, BMI<25, and BMI>25). 

Figure 2: Please include the definition of * and # in the figure legend.

**Conclusions**

-Are the conclusions supported by the data presented?

-Are the limitations of analysis clearly described?

-Do the authors discuss how these data can be helpful to advance our understanding of the topic under study?

-Is public health relevance addressed?

Reviewer #1: The conclusions are supported by the data and the authors have been clear about the limitations of the data. The authors discuss well the relevance of the data to human health as it relates to lipid levels in individuals with high BMI. Of course it is a big jump from this study to proposing Sh infection would have similar effects on Dutch people who have never previously encountered Sh infection.

Reviewer #2: The authors have interpreted their results and limitations well, highlighting where their findings fit with the (rapidly growing) literature in this field relating to worms and metabolism. 

There remains obvious mechanistic unknowns as to HOW Sh regulates metabolism, some more in depth analysis of cytokines, cellular immune responses and the microbiome would have been useful to understand this, but this could form the basis of future work (and is well discussed anyway).

Reviewer #3: I recommend the authors to include a short conclusion paragraph at the end of the discussion. 

Paragraph 273-292: The article by Cortes-Selva D et al. (Frontiers in Immunology 12;9:2580) should be included since it reinforces the hypothesis that Schistosoma induced Th2 response confers protection from hyperlipidemia, atherosclerosis, and glucose intolerance.

**Editorial and Data Presentation Modifications?**

Reviewer #1: The 'Statistical Analysis' paragraph is well written. These are a good choice of tests, consideration of confounding factors, and multiple testing corrections. 

Tables S1 & S2 is where they show the raw Odds Ratio and confounder adjusted OR for egg and serum levels. Tables S1, S2 are very informative, and are well presented by showing both raw and adjusted results (that consider multiple important confounding factors). 

Tables S3, S4, S5 are mislabelled. Table S3 summarises factors stratified by CAA range; the Eosinophil response appears to show incredibly strong correlation to stratification level, it is unfortunate they lost some measurements for Eosinophils as stated, but their choice of statistical test is well suited to different sample sizes. Table S4 shows the disparity between gender between BMI >/< 25 groups, showing the importance of showing adjusted OR – It is important that the presentation of raw data is included.

Fig. 1 It is interesting when whole population the HDL-C has a significant p-value but when you split it into lean & obese there is no significant p-value in either, despite the same trend. The overall trend is clear however that serum CAA levels are associated with many cholesterol measurements in the obese category. 

Fig. 2 a and b are skillfully plotted - and a statement is needed with respect to what p-values are associated to (#/*/#*) on the heatmap.

Reviewer #2: (No Response)

Reviewer #3: As mentioned above, the tables presented in the main text should be modified to include the CAA data. CAA text is more sensitive and should be considered as the "true positive and true negatives." The analysis based only on the direct observation of eggs in urine does not need to be included in the main text since it contains false negatives. 

QUICKI should be calculated and included in the paper.

**Summary and General Comments**

Reviewer #1: This is an interesting and provocative study. I do think it would be interesting to put in a comparison of all the individual immune responses to Sh (as measured by eosinophil levels) and the effect on lipid levels.

Reviewer #2: I think this is a timely, novel and interesting study. Some clarification as to the relative importance of S hematobium versus the other co-endemic helminths would add important supportive evidence

Reviewer #3: Zinsou et al. observed that overweight/obese individuals infected with Schistosoma haematobium have an improved lipid profile. This manuscript is relevant to the readers of PlosNTDs, and this study is the first report that evaluated the impact of a helminth infection in overweight/obese individuals from endemic areas. The main limitation of the study is the sample size. The authors need to inform how the sample size was calculated. The manuscript is well written, but the data could be better presented. The Sh+ and Sh- groups should be divided based on CAA test since it is more sensitive than the parasitological exam (presence of eggs in urine). However, the authors present this data mostly as supplemental material.

PLOS authors have the option to publish the peer review history of their article (what does this mean?). If published, this will include your full peer review and any attached files.

Reviewer #1: No

Reviewer #2: No

Reviewer #3: No
---

## [Editor Report · Decision Letter 1]

8 Jun 2020

Dear Dr. Guigas,

We are pleased to inform you that your manuscript 'Schistosoma haematobium infection is associated with lower serum cholesterol levels and improved lipid profile in overweight/obese individuals' has been provisionally accepted for publication in PLOS Neglected Tropical Diseases.

Best regards,

Keke Fairfax, PhD

Deputy Editor

Michael Hsieh

Deputy Editor

Your revisions have addressed the concerns raised by the reviewers.

---

## [Editor Report · Acceptance letter]

24 Jun 2020

Dear Dr. Guigas,

We are delighted to inform you that your manuscript, "Schistosoma haematobium infection is associated with lower serum cholesterol levels and improved lipid profile in overweight/obese individuals," has been formally accepted for publication in PLOS Neglected Tropical Diseases.

Best regards,

Shaden Kamhawi

co-Editor-in-Chief

Paul Brindley

co-Editor-in-Chief
